# The Impact of Social Isolation on the Subjective Well-Being of Older People in China: An Empirical Analysis Based on the 2021 China General Social Survey

**DOI:** 10.3390/ijerph22040501

**Published:** 2025-03-26

**Authors:** Keikoh Ryu, Zaiqing Chen

**Affiliations:** Graduate School of Business Administration, Keio University, Yokohama 2238526, Japan; chenzq182931@keio.jp

**Keywords:** social isolation, self-assessment of health, anxiety about old age, insurance participation, subjective well-being

## Abstract

This study focuses on the psychological state and economic preparedness of socially isolated older individuals in China’s rapidly aging society. Both a simple model and an extended model were developed and tested to explore the impact of social isolation on the subjective well-being of older adults by analyzing how social isolation, self-assessment of health, and anxiety about old age affect subjective well-being. The results indicated that both social isolation and self-assessment of health have a strong influence on subjective well-being, and that social isolation significantly alters older individuals’ perception of caregiving responsibility. While private insurance participation had a significant impact on the well-being of younger individuals, its effect on older adults was limited. These findings provide valuable insights for improving support systems for older people.

## 1. Introduction

In recent years, with the development of modern society, China has become a country with a sizable aging population. As it develops support systems and frameworks for older adults, problems have become increasingly apparent [1]. According to the 2024 China Health and Retirement Longitudinal Study [2], by the end of 2023, the number of people aged ≥60 in China totaled 297 million, accounting for 21.1% of the total population. Further estimates based on this study indicate that 54% of people aged ≥60 live independently, exceeding the proportion of those living with children since 2018, which marked a historic turning point [3]. This rapidly aging society raises questions about the psychological state and economic preparedness of older adults living alone.

Previous studies have summarized various aspects of older adults’ isolation in modern China, including their lack of mental and economic preparedness, insufficient participation in social security, and heavy reliance on family care—particularly among the one-child generation of empty nesters in rural areas (as their children leave to work in cities) [4]. Essentially, this research suggests that the underlying issue is not the fact that these older people are living alone, but rather the social isolation they experience, which holds a number of risks [5].

As part of an effort to identify ways to mitigate these risks, this study aims to explore the determinants of subjective well-being among older individuals living in isolation in order to better understand their needs and improve their quality of life. Specifically, it attempts to establish and test a model of how social isolation impacts the subjective well-being of older adults, so as to develop more effective policies and practices with respect to enhancing social security and developing support systems in an increasingly aging society.

### 1.1. Structure and Analytical Method of the Study

This paper is divided into three sections (Figure 1). The first section discusses the development of a simple analytical framework to examine the effects of social isolation on older individuals’ self-assessed health status and subjective well-being. The second section builds upon the results of the first to develop an extended model that explores the broader effects of social isolation on the subjective well-being of older adults, clarifying the impact pathways of several factors through which this process occurs. The third section discusses the resulting data with a particular focus on gender and other key factors. The underlying aim of this approach is to provide preliminary empirical insights into the specific challenges posed by China’s aging society, including gender differences in psychological states, insurance coverage, and the evolving recognition of caregiving responsibilities.

### 1.2. Data

This study utilizes data from the 2021 Chinese General Social Survey (CGSS), a comprehensive and longstanding national survey jointly conducted by the China Survey & Data Center of Renmin University of China and the Hong Kong University of Science and Technology. Details are provided in the China General Social Survey Report (2003–2008) [6], and information on the progress of the project, data usage, and research outcomes, etc., can be found on the Renmin University of China website (http://cgss.ruc.edu.cn/, accessed 10 October 2024). Additionally, due to the impact of the COVID-19 pandemic, survey data from 2020 and 2022 have not yet been made publicly available. Though more recent data have not yet been released, the findings derived from this analysis offer valuable insights into the status of older adults in China, and may be used as a reference for gaining a better understanding of the effects of the aging-related policies implemented since 2017.

## 2. The Impact of Social Isolation on Self-Assessed Health Status and the Subjective Well-Being of Older Adults

### 2.1. Prior Research

#### 2.1.1. The Challenges Facing China’s Aging Society

As a developing country, China is still in the early stages of recognizing and addressing the myriad challenges posed by an aging society. However, the pace of aging across the country is rapid, and the resulting problems have been compounded by the widespread phenomenon of “getting old before getting rich”. This refers to the phenomenon in which a society’s population enters old age before achieving economic prosperity or social development, leading to a decline in economic growth due to the resulting imbalance in the population structure (pp. 19–20) [7,8].

In 2017, the Chinese government established a framework for providing support services for older adults in its *13th Five-Year Plan for the Development of Elderly Services and Construction of the Elderly Care System*, which was itself based on related policy directives issued in 2011. This framework proposed “home-based elderly care as the foundation, community-based elderly care as the support, and institutional elderly care as the supplement” *“The 13th Five-Year Plan” for the Development of National Elderly Affairs and the Construction of the Pension System*). However, notable challenges remain, particularly given the erosion of family caregiving due to changes in intergenerational household relationships and the one-child policy. The one-child policy was a birth control policy implemented by China between 1979 and 2014, which limited couples to having one child per generation. On 29 October 2014, during the Fifth Plenary Session of the 18th Central Committee, the policy was abolished as one of the measures to address the country’s aging society. At the same time, social security systems and community-based support structures, which rely on communication and mutual assistance between residents, including older adults, remain underdeveloped (pp. 4–5) [1].

In addition to the issues associated with social isolation among older adults, including solitary deaths and suicides, the aging of China’s society has also increased existing pension and healthcare burdens [9]. In a study on the caregiving challenges for the parent generation of one-child families, whose social isolation tends to be particularly pronounced as a result of the one-child policy [10], found that this generation has not adequately prepared themselves mentally, psychologically, or financially for their care and retirement needs. They also found that China’s social security system remains inadequate and incomplete, and that parents with only one child tend to face difficulties when it comes to caregiving choices. Whether they choose to move into their child’s home, enter a care facility, or live alone, none of these options provides an easy solution to the caregiving challenges faced by this generation.

#### 2.1.2. The Impact of Social Isolation on Attitudes Toward Self-Assessed Health and Subjective Well-Being in Older Adults

It is well-established that humans have a fundamental need to connect with others. In particular, individuals have a “need to belong”, characterized by a desire to join social groups and be accepted by others [11]. When this basic need goes unmet, it can lead to social isolation, triggering complex behavioral responses [11]. Moreover, the incidence of social isolation among older individuals living alone (42.6%) is significantly higher than among those living with others (26.6%) [12].

As for the definition of “social isolation”, various approaches have been taken. Some research understands social isolation as “a state of having minimal contact with family or community” [13]. Another nationwide survey by Japan’s Naikaku-kanbō on the conditions of loneliness and isolation (2023) [14] assessed social isolation based on the frequency of communication with non-cohabitating family and friends (social interactions), face-to-face conversations, phone calls (including mobile communication), face-to-face communication with cohabitating individuals, participation in social activities, and the extent of support from family or friends in times of anxiety or concern.

Moreover, as social isolation among older adults intensifies, mental health problems are growing more severe and widespread. According to the China Elderly Mental Health Report issued in 2018, 63% of older adults in China report feeling lonely on a daily basis, highlighting the impact of social isolation on their well-being. Okamoto [15] has also found that individuals with insufficient social interaction experience a marked decline in subjective well-being, and are more likely to avoid activities involving social support. Conversely, it was found that maintaining a good level of social interaction improves subjective well-being, and leads to a more active lifestyle that helps counteract the need for caregiving services.

The concept of “subjective well-being” has also gained increasing attention as an important indicator for analyzing individual psychological states in fields such as policy evaluation and sociology [15]. It essentially refers to an individual’s overall evaluation of their life, including satisfaction with life, optimism about the future, and assessment of personal life experiences [15]. As a psychological indicator, it has been used in numerous gerontological studies to assess the psychological health and attitudes of older individuals [16].

Several factors impacting the subjective well-being of older adults have been analyzed from various angles. Luo et al. [12], for example, conducted a meta-analysis examining the social isolation of older individuals in Chinese communities and its attendant risk factors, which found that the prevalence of social isolation among older community residents reached 29.5%, depending on such factors as gender, advanced age, educational background, self-assessed health, marital status, lack of family support, and low levels of social participation.

“Self-assessed health” is also a critical factor affecting the subjective well-being of older individuals living alone. According to Li et al., older individuals living alone who rated their health as “average” or “poor” had a 1.94 times and 2.25 times higher risk of feeling lonely, respectively, compared to those who rated their health as “good”. In addition, older individuals who were at least partially unable to live independently had a 1.9 times higher risk of feeling lonely compared to those who were self-sufficient.

A lack of social support and a sense of isolation can cause psychological stress, which exacerbates anxiety about old age as well as stress due to physical health issues and financial insecurity, and can lead to a vicious cycle which serves to further deepen isolation [17]. However, further research on the impact of isolation on the psychological health and well-being of older individuals is necessary [18].

As set forth in Figure 2 below, and based on the above findings, three hypotheses were proposed:

**H1:** 
*Social isolation has a significant impact on subjective well-being, with higher levels of isolation leading to lower subjective well-being;*


**H1a:** 
*Social isolation significantly impacts the self-assessed health of old*
*er individuals, with higher levels of isolation leading to lower self-assessed health; and*


**H1b:** 
*Social isolation affects self-assessed health, which, in turn, influences subjective well-being.*


### 2.2. Data Utilization and Analytical Method

#### 2.2.1. Data Utilization

A quantitative approach was taken to verify the above-referenced model, using data from the 2021 CGSS to conduct a large-scale sample data analysis to capture a more realistic representation of current social conditions. Specifically, this analysis focused on individuals aged ≥60 as of 2021, with respondents under 60 years of age (1707 cases) grouped as a comparative cohort. To improve accuracy, the data also underwent cleaning procedures, including checking for missing values, handling outliers, and correcting input errors. Crucial variables which rendered data invalid were also addressed, resulting in a total of 1010 valid responses from individuals aged ≥60 for further analysis.

#### 2.2.2. Variable Settings

For the purposes of this study, the dependent variable for the overall model (subjective well-being) was derived from the CGSS question, “All things considered, do you feel happy with your life?”, with the respondents’ answers used to reflect subjective well-being. Data involving self-assessed health were derived from the answer to the following question: “All things considered, how would you rate your health?”.

The independent variable—social isolation—was constructed by performing a principal component analysis (PCA) on responses to the following questions: “How often do you interact with your neighbors?”, “How often do you interact with relatives who do not live with you?”, “How often do you interact with your friends?”, “How often do you interact with other acquaintances?”, and “Do you have someone to talk to about your worries?” Based on the results of the PCA, these questions were then weighted to create a composite variable for social isolation. A detailed description of the variables is provided in the section below (Table 1).

The Cronbach’s alpha coefficient for the standardized five items of social isolation was 0.654, which, though slightly low, is still considered acceptable. The Kaiser–Meyer–Olkin (KMO) test and Bartlett’s test of sphericity were then used to confirm the validity of the data. Although the overall KMO value of 0.654 is on the lower end of the acceptable range, it still exceeds the threshold of 0.6, suggesting that the data are suitable for factor analysis.

Given that the sensitivity analysis showed no significant changes in the KMO value after removing any individual variable, the data were considered stable and robust for further analysis. There was also some concern that replacing the data or making major adjustments could potentially alter the focus of the research and introduce unnecessary complexity. Therefore, since the resulting dataset still captured the research objectives effectively, we proceeded with the analysis without any major modifications.

The results of Bartlett’s test of sphericity (*p* < 0.05) further confirmed that the data were suitable for factor analysis. Specifically, since the variance explained by the factors extracted after rotation was 35.858% and 16.747%, respectively, with a cumulative variance explained rate of 52.606% after rotation (which meets the general criteria for factor analysis), the extracted factor was used to represent the independent variable “social isolation”.

#### 2.2.3. Model Analysis

Regression analysis was conducted to test the two pathways of the model, yielding the following results (Table 2.):

Our analysis of the mediating effect of self-assessed health on the relationship between social isolation and subjective well-being showed significant results. The total effect (c) was −0.105, confirming that social isolation reduces subjective well-being significantly. The indirect effect (a × b) was −0.026, and the bootstrap confidence interval (−0.049, −0.016) suggested that the mediating effect was significant. The direct effect (c′) was also significant at −0.079, indicating that social isolation has a direct negative impact on subjective well-being, independent of self-assessed health.

These results further suggest that social isolation lowers self-assessed health significantly (a = −0.191), and that an improvement in self-assessed health contributes to an increase in subjective well-being (b = 0.135). Overall, it was confirmed that self-assessed health functions as an important mediating variable in the relationship between social isolation and subjective well-being.

### 2.3. Summary

The regression analysis confirmed that self-assessed health tends to worsen as social isolation increases, and that self-assessed health has a significant positive impact on subjective well-being, suggesting that a higher value of self-assessed health leads to improved subjective well-being. These results also indicate that self-assessed health plays a mediating role in the impact of social isolation on subjective well-being, thereby shedding light on one of the mechanisms through which social isolation reduces the subjective well-being of older adults. The proposed estimation model was thereby validated.

## 3. An Extended Model of the Impact of Social Isolation on Subjective Well-Being in Older Adults

To meet the diverse needs of older individuals, it is increasingly necessary to establish a comprehensive system of support services. This section draws on the results of prior research to extend the model discussed in Section II by incorporating additional factors, including anxiety about life in old age and perceptions of caregiving responsibilities, in an effort to analyze how these variables mediate the relationship between social isolation and subjective well-being in older adults.

The challenges of an aging society and the caregiving responsibility dilemma are closely associated with the changes in family structures that have emerged in modern society. In traditional societies, the responsibility for supporting and caring for older adults tended to fall upon the younger generation. China has historically emphasized a vertical parent–child relationship, which dictates that family members, particularly in rural areas, support their elders in later life [19]. In today’s aging society, however, this balance has been disrupted, with extended parental longevity leading to prolonged parent–child relationships and the aging of the children themselves, causing a significant decline in available family caregivers as well as intergenerational conflict [20].

Moreover, due to shifts in the composition of family households, as individuals age, they tend to become increasingly isolated from family and social relationships, leading some to experience psychological issues associated with loneliness, depression, anxiety, and reduced self-esteem [20]. As a result, caregiving in contemporary society has shifted from an intimate, family-bound activity to a more externalized, socialized act [4].

### 3.1. The Extended Estimation Model

Given the above, it was hypothesized that social isolation impacts older individuals’ attitudes toward subjective well-being through four pathways, the first of which involves a direct route in which social isolation directly affects subjective well-being. In other words, older individuals who experience severe social isolation are expected to report lower self-assessed health and subjective well-being.

The other three pathways are mediating routes, in which social isolation influences subjective well-being indirectly through its impact on self-assessed health, anxiety about old age, and perceptions of caregiving responsibility, each of which in turn contributes to shaping subjective well-being in the context of social isolation.

Therefore, as set forth in Figure 3 below, the following hypotheses were proposed:

**H1:** 
*Social isolation affects subjective well-being significantly, with higher levels of isolation leading to lower subjective well-being through self-assessed health;*


**H2:** 
*Self-assessed health influences anxiety about old age significantly, with poorer health leading to more anxiety, which, in turn, affects subjective well-being;*


**H3:** 
*Social isolation influences anxiety about old age through self-assessed health, which, in turn, affects subjective well-being;*


**H3a:** 
*Social isolation influences subjective well-being through self-assessed health and anxiety about old age; and*


**H4:** 
*Self-assessed health influences subjective well-being both directly and indirectly through anxiety about old age.*


### 3.2. Analytical Method

Ideally, structural equation modeling (SEM) would have been used to construct and test the extended model, but the fit indices for the underlying data did not satisfy the required criteria for SEM. Instead, the model was verified through linear regression analysis, as described below:(1)Group-based regression analysis: separate analyses were conducted for older and younger groups, with heterogeneity testing performed to identify differences between the models for each group. Based on the results, each factor’s impact on subjective well-being in older adults was then assessed.(2)Pathway-specific analysis: the specific pathways for each hypothesis were analyzed, and a final extended model was proposed based on the resulting findings.

#### Data Utilization and Explanation of Additional Variables

As with the simple model discussed above, the extended estimation model was also based on data from the 2021 CGSS but includes two additional variables: “anxiety about old age”, derived from three survey questions—“Do you live alone?”, “Are you worried about your future life?”, and “Do you feel secure about your future?”—and “Perception of Caregiving Responsibility”, as reflected in the question, “For elderly people with children, who do you think should primarily be responsible for caregiving?”

To verify that these items consistently measure the underlying concepts, internal reliability and validity were assessed. The three questions related to “anxiety about old age” achieved a Cronbach’s alpha coefficient of 0.807, indicating good reliability. The corrected items’ total correlation values were also all above 0.4, suggesting strong correlations between the items and high reliability. Additionally, the KMO test and Bartlett’s test were conducted to assess validity, with a KMO value of 0.705, indicating that the data had good validity and were suitable for information extraction.

An explanation of these additional variables and their corresponding values can be found in Table 3 below.

### 3.3. Model Regression Analysis

To validate the model, multiple regression analysis was conducted by entering the independent variables—living alone, degree of social isolation, participation in private insurance, anxiety about old age, and self-assessed health—with subjective well-being set as the dependent variable.

#### 3.3.1. Multicollinearity Testing

Since multicollinearity may render the model’s estimation results unstable due to high correlations between independent variables, collinearity was analyzed before conducting regression analysis. The absolute values of correlation coefficients between the six main variables were all below 0.8. In addition, variance inflation factors (VIFs) were calculated to assess collinearity, with a VIF threshold of 5. Notably, none of the VIF values exceeded 5 and tolerance levels were ≥0.1, thereby confirming that multicollinearity was not present among the variables.

#### 3.3.2. Heterogeneity Testing

A heterogeneity test was also conducted by dividing the sample into older and younger groups. This allowed for an examination of the relationship structure between variables and the model fit for each group, including an evaluation of how the predictor variables impacted the dependent variable for each group, thereby providing a more detailed understanding of the characteristics of older individuals and the factors influencing their subjective well-being.

The analysis results are presented in Table 4 and Table 5. Chow test results (*p* < 0.05) confirmed structural differences between the models for older and younger groups, indicating that age (specifically, whether an individual is older) brings about a structural change in the determinants of subjective well-being. Comparing values, the explanatory power of the younger model was lower, whereas the overall model and the older model showed relatively better precision.

In the overall model and the older group, living alone did not affect subjective well-being significantly. In the younger subgroup, however, living alone did have a significant negative effect on subjective well-being (b = −0.076, *p* < 0.01), suggesting that living alone reduces subjective well-being only among younger adults.

As for the degree of social isolation, both the overall model and the subgroup models showed a significant negative impact on subjective well-being, with effects being similar across both older and younger groups (*p* > 0.05). This suggests that social isolation lowers the levels of subjective well-being in both age groups.

Anxiety about old age also significantly affected subjective well-being in both groups, with no notable difference between the two (*p* = 0.243), implying that higher anxiety about old age tends to correlate with lower subjective well-being across both groups.

For self-assessed health, however, a significant difference was observed between older and younger adults. In the younger group, self-assessed health had a stronger positive impact on subjective well-being (b = 0.198, *p* < 0.01), suggesting that health perception has a greater impact on well-being among younger individuals but weakens slightly in older individuals.

Finally, participation in private insurance was significantly associated with subjective well-being in both the overall model and the younger group (b = −0.098, *p* < 0.01), but not in the older group. This indicates that while private insurance participation may enhance subjective well-being for younger individuals, this is not the case for older adults.

#### 3.3.3. Mediation Effect Analysis

Based on the results of the multiple regression analysis, verification of each mediating effect was conducted through further regression analysis to obtain insights that could improve and refine the extended model.

First, the hypothesis that self-assessed health influences subjective well-being through anxiety about old age (Table 6) was tested. The results confirmed a partial mediation effect, indicating that improvements in self-assessed health reduce anxiety about old age, which in turn enhances subjective well-being. This finding highlights the significant role of anxiety about old age in shaping subjective well-being, demonstrating that anxiety about old age partially mediates the relationship between self-assessed health and subjective well-being. Furthermore, it was also confirmed that self-assessed health exerts a direct effect on subjective well-being, suggesting that the relationship between self-assessed health and subjective well-being is only partially mediated by anxiety about old age.

The hypothesis of social isolation ⟶ caregiving responsibility attribution ⟶ subjective well-being (Table 7) was tested next. The indirect effect mediated by caregiving responsibility attribution (a*b) was found to be −0.001, which was not statistically significant (*p* = 0.888, z = −0.141). This indicates that while social isolation negatively impacts subjective well-being, this effect was not mediated by caregiving responsibility attribution.

The results of the hypothesis of social isolation ⟶ self-assessed health ⟶ anxiety about old age (Table 8) yielded a full mediation effect, however, suggesting that the impact of social isolation on anxiety about old age is fully mediated through self-assessed health. In other words, social isolation influences self-assessed health, which, in turn, affects anxiety about old age. Moreover, since the direct effect was insignificant, it can be concluded that the impact of social isolation on anxiety about old age is entirely mediated by self-assessed health.

### 3.4. Final Extended Model

Based on the results of the above analysis, a final extended model was constructed as shown in Figure 4 below:

As discussed above, since subjective well-being was found to differ depending on differences in the recognition of caregiving responsibility attribution, the mediating effect route of caregiving responsibility attribution was removed from the final model.

## 4. Analysis of Gender Differences in Older Adults’ Psychological Well-Being, Insurance Participation, and Shifting Recognition of Caregiving Responsibilities in China

### 4.1. Prior Research

#### 4.1.1. Gender Differences in the Psychological State of Older Adults

The results discussed above suggest that in order to gain a deeper understanding of the psychological well-being of older individuals at the personal level, it is also necessary to consider gender differences.

Extensive research has been conducted on gender differences in the psychological state of older adults. Studies by Luo et al. [12] revealed that the incidence of social isolation among older men in Chinese communities is slightly higher than that of older women, largely due to such factors as changes in living environments, including retirement. In a study by Matsuura and Ma [5], a comparison of the subjective well-being of older men and women living alone in Japan showed that living alone increased the happiness of women but decreased that of men. In China, however, while living alone continued to have a negative impact on men’s happiness, it had no significant impact on women [5].

Established more than 175 years ago, PCA Life Assurance provides life insurance, health insurance, asset management, and various other services (Prudential Life Insurance website (https://www.pcalife.com.tw/zh/about-prudential/; accessed 11 October 2024). A 2023 survey by PCA Life Assurance found that women’s post-retirement anxieties exceeded those of men in many areas, due in part to the fact that women typically live longer than men and were therefore more concerned about their future. This is consistent with the results of a 2014 report concerning China’s aging problem, which found that women are more likely to experience loneliness and depression as they age, which may exacerbate these anxieties.

Based on this research, the following hypotheses were proposed:

**H2a:** 
*There is a gender difference in social isolation and subjective well-being among old*
*er individuals, with older men experiencing higher levels of well-being;*


**H2b:** 
*There is a gender difference in anxiety about life in old age, with old*
*er women experiencing higher levels of anxiety.*


#### 4.1.2. The Impact of Social Isolation and Insurance Participation on the Elderly's Psychological State

Since China began reforming its medical security system in 1998, it has strengthened basic pension insurance (the first pillar) while actively developing corporate and occupational pensions (the second pillar) and personal insurance (the third pillar) in an effort to build a more diversified pension system (National Healthcare Security Administration, 2023) [21]. The third pillar also includes social pension insurance and medical insurance, which serve as a support system for older adults, complementing the basic pension from the first pillar. This multi-pillar system is aimed at improving the quality of life for citizens in their later years [22].

However, there are several outstanding issues. For one, the social pension and medical insurance systems are operated separately, instead of being integrated into a unified framework [23]. Data have also shown that the rate of personal insurance uptake is low, and that the number of individuals actively paying into their basic pension insurance accounts, particularly in rural areas, remains small [22]. The imbalance in service provisions, limited options available, and complexity of enrollment procedures only contribute to the problem [22].

In fact, research suggests that active participation in both social basic insurance and private insurance significantly increases subjective well-being [24] also found that social basic insurance can shift the perception of caregiving responsibility for older individuals from being predominantly family-based to including both individuals and the government as responsible parties.

Based on these findings, the following hypothesis was proposed:

**H3:** 
*Social isolation significantly affects insurance participation, with higher levels of isolation leading to a lower likelihood of insurance participation.*


#### 4.1.3. Changes in Social Isolation and Perceptions of Caregiving Responsibility

A study on older individuals in China by Zhang et al. [25] further suggests that intergenerational family support can enhance older individuals’ sense of purpose and subjective satisfaction, but that the degree of this impact varies depending on the individual’s perceived social status [25]. These changes in the family and surrounding environment also directly influence attitudes toward social support measures, including insurance enrollment and caregiving responsibility.

Other studies have found that the more children an older individual has, the less demand there is for social support services [26]. In addition to physical and cognitive decline, however, social isolation and a reluctance to acknowledge disconnection from society has also been found to contribute to a dismissive attitude toward social security and support [27]. Ultimately, in order to make social support services more accessible to older individuals living in isolation, it is first necessary to address traditional views on caregiving responsibility which dictate that it should rest solely with the family [26].

Based on these findings, the following hypothesis was proposed:

**H1:** 
*The more severe an old*
*er individual’s social isolation, the stronger their tendency to believe caregiving is a family responsibility.*


## 4.2. Analytical Method

Descriptive statistics were used to examine respondents’ insurance participation and caregiving responsibility attribution, with Chi-square tests and *t*-tests conducted to assess differences across groups. The Chi-square test was chosen because it is suitable for analyzing relationships between categorical variables, making it effective in revealing distributional differences across groups.

Multiple logistic regression analysis was used to analyze how social isolation conditions impact older individuals’ perceptions of caregiving responsibility, as it is well-suited for situations involving binary dependent variables and allows for the quantification of the effects of independent variables on the outcome variable while controlling for other factors. By comparing different levels of social isolation, the study examines how varying degrees of isolation influence older individuals’ perceptions of who should be responsible for their care.

Moreover, given that this study is focused on the social isolation of older adults in China, and that living alone is strongly associated with social isolation among this age group, older adults were also categorized into those living alone and those not living alone.

## 4.3. Data Analysis

### 4.3.1. Analysis of Insurance Participation

The hypotheses regarding insurance participation status and various other indicators were evaluated using Chi-square tests and *t*-tests to assess relationships and differences in the data.

First, the insurance participation status of older and younger groups was compared. As outlined in Table 9 below, Chi-square tests on insurance participation by urban/rural basic pension insurance and two types of private insurance showed that younger participants were significantly more likely to participate than older participants. Among those aged ≥60, the effect of living alone on insurance participation was also analyzed, indicating that living alone significantly influenced participation in one category—urban basic medical insurance/new rural cooperative medical insurance/public healthcare (*p* < 0.05). However, living alone showed no significant impact (*p* > 0.05) on participation in urban/ rural basic pension insurance or private medical insurance, suggesting that living situation does not influence participation in these types of insurance (Table 9 and Table 10).

In order to compare anxiety about old age, self-rated health, and subjective well-being by gender for individuals aged ≥60, a *t*-test was conducted. For anxiety about old age, the results indicated that women (*n* = 1503, M = 2.58, SD = 1.00) reported significantly lower levels than men (*n* = 1426, M = 2.74, SD = 1.04), suggesting that women tend to feel less anxious than men about growing older (T = 2.423, *p* = 0.016) (Table 11).

For self-rated health, men tended to perceive their own health as better than women. And for subjective well-being, men (M = 4.11, SD = 0.81) scored significantly higher than women (M = 4.01, SD = 0.90) (T = 2.651, *p* = 0.008), suggesting that men feel a stronger sense of subjective well-being than women. These results support both hypotheses H2a and H2b.

#### 4.3.2. Multinomial Logistic Regression Analysis on the Influence of Social Isolation on Older Adults’ Perceptions of Care Responsibility

A multinomial logistic regression analysis was then conducted to examine the impact of social isolation on older individuals’ perceptions of caregiving responsibilities, in order to clarify how the degree of social isolation is related to the recognition of caregiving responsibilities among older adults. Specifically, the study explored how older individuals with stronger feelings of isolation perceive their caregiving responsibilities, and how this perception affects their relationships with others and society at large.

For the purposes of this analysis, the category of “mainly relying on children” was chosen as a reference point, with the aim of evaluating changes in the likelihood of selecting other categories in relation to it. This approach allowed for a clearer understanding of the extent to which each category could be chosen compared to “mainly relying on children”, thereby offering important insights into the impact of social isolation and the differences in recognition of caregiving responsibilities.

Results indicated that for the perception of “relying on the government”, the regression coefficient for social isolation was −0.095, with a *p* value of 0.318, showing no statistically significant effect. The odds ratio (OR) was 0.909, with a 95% confidence interval (CI) of 0.755–1.096, suggesting that stronger feelings of isolation have a minimal impact on the perception of relying on the government. Similarly, for the “self-responsibility” perception, the regression coefficient was 0.063 with a *p* value of 0.534, again indicating no significant effect. The OR was 1.065, with a 95% CI of 0.872–1.301, suggesting that the influence of isolation on the recognition of self-responsibility is also weak.

By contrast, the analysis of “shared responsibility among three parties” showed a regression coefficient of −0.333 with a *p* value of 0.000, which is statistically significant. The OR was 0.717, with a 95% CI of 0.618–0.831, suggesting that as social isolation increases, the recognition of shared responsibility among three parties decreases. While the impact of social isolation on perceptions of caregiving responsibilities varied by category, a clear relationship was observed with respect to “shared responsibility among three parties”.

Therefore, as shown in Table 12 above, individuals with stronger feelings of social isolation are more likely to select the option “mainly relies on children”, confirming the validity of hypothesis H4.

As shown in Table 13 based on the above multinomial logistic regression analysis, an evaluation of the differences in subjective well-being among older individuals with varying perceptions of caregiving responsibility revealed an overall *p* value of <0.05, indicating that the allocation of caregiving responsibility has a significant impact on subjective well-being and suggesting that self-determination may in fact enhance well-being.

### 4.4. Summary

Several insights can be gleaned from the above analysis. First, an improvement in the coverage levels of the three pillars of retirement insurance can strengthen the stability of retirement security, particularly for younger individuals. It was also found that while older men tend to experience a stronger sense of subjective well-being than women, they also harbor greater anxiety about old age. As for perceptions of caregiving responsibility, older individuals with a higher degree of isolation are more likely to believe that they should rely on children for their care. Finally, it was found that the stronger the perception that caregiving should be the responsibility of children, the lower the level of subjective well-being.

## 5. Discussion

### 5.1. Conclusions

The purpose of this study was to examine a number of significant issues facing China’s aging society, including social isolation, anxiety about old age, subjective well-being, perceptions of caregiving responsibility, and insufficient insurance participation, with a focus on identifying the pathways through which these various factors influence one another. Using large-scale survey data from China, both a simple model and an extended model were proposed and analyzed, leading to several conclusions.

First, social isolation significantly affects older individuals’ subjective well-being as well as self-assessed health and anxiety about old age. Older adults who perceive their health negatively tend to experience increased anxiety about the future, which, when combined with social isolation, serves to worsen their well-being. Promoting social connections and supporting older individuals in cultivating a positive perception of their health are essential for addressing these issues.

Second, older individuals with increased feelings of social isolation are more likely to believe that caregiving should primarily be the responsibility of their children, which may be attributed to a combination of psychological, social, and cultural factors. As they age, socially isolated individuals gradually lose social connections, often experiencing loneliness and emotional loss [28]. This intensifies their desire for family support, leading them to seek emotional and practical assistance from their children. This lack of social resources can also cause older individuals to feel that their societal role and value have diminished, further reinforcing their reliance on traditional familial support.

Third, participation in private insurance has a significant effect on subjective well-being among the general population and younger participants, though not older individuals. While private insurance enrollment rates remain low among older adults, they are gradually increasing among the younger population.

Given these results, it is essential to introduce programs that strengthen social connections so as to reduce feelings of social isolation among older adults. Helping older adults view their health positively through regular health check-ups and health management programs is also crucial. Finally, awareness campaigns to promote the use of caregiving services provided by local communities and the government are indispensable in emphasizing the importance of support systems beyond the family, and strengthening industrial support and collaboration for the care of older adults.

### 5.2. Contributions to Research and Practice

While the focus of this study is to explore how social isolation impacts subjective well-being, anxiety about aging, and the attribution of caregiving responsibility from the perspective of older individuals, thereby deepening our understanding of older adults’ quality of life and mental health, it also has important theoretical and practical implications for the formulation of policies and intervention strategies tailored to different age groups.

While numerous studies have addressed social isolation and subjective well-being among older adults, quantitative research on older individuals in China remains relatively limited and is often biased. However, interest in aging (healthy aging in particular) in China is increasing overall, with the number of English and other international publications outnumbering those in Chinese and focused generally on policy support, health services, and public health [29]. Significant differences can also be found in the focus and methodologies of Chinese and international studies.

Building on prior research, this paper utilizes large-scale survey data from China to conduct a quantitative analysis in order to provide more universal and objective insights. It aims to strengthen the foundation of research on older adults in China by presenting a novel perspective on the country’s aging population with a particular focus on the impact of social isolation on the attribution of caregiving responsibility among older adults.

Its analysis of numerous other factors, including social isolation, low insurance participation rates, and attitudes toward caregiving among older individuals in China, also holds valuable insights for enhancing awareness of the importance of non-family and societal support systems. This study should therefore provide a meaningful foundation for future research on China’s aging society, while supporting the development of effective policies and interventions tailored to various age groups.

### 5.3. Limitations and Future Directions

That said, there are several limitations. First, with respect to insurance participation by older adults, the analysis is relatively superficial. In fact, insurance enrollment rates in China are closely tied to a number of economic conditions and regional factors, with many areas experiencing a decline in enrollment between 2010 and 2020 [30]. This study also does not account for differences in income levels or regional variations (e.g., between urban and rural areas), which would be critical for providing a more comprehensive understanding. Future research should therefore address these differences, so as to provide a more precise reference for tailoring support services to different subgroups of older adults.

While this study offers important insights into the general population’s circumstances, a deeper understanding of individual participants’ actual circumstances is also needed to further substantiate its conclusions. Combining validated quantitative measures with qualitative approaches could provide a more robust foundation for future research, which will be necessary to offer more specific insights into the most effective way to address the myriad social issues facing older adults in China and elsewhere.

Moreover, as a preliminary exploration of the issue, this study relies primarily on prior research and available third-party data, from which it has selected variables suitable for its framework. Therefore, while there are certainly broader considerations to account for, the chosen variables were selected based on the data availability constraints. Future studies should therefore expand on this foundation by accounting for a more extensive and diverse range of variables.

Finally, this study does not consider how older adults’ quality of life, mental health, subjective well-being, support access, and insurance participation has been affected by the COVID-19 pandemic, which should be prioritized in future research on the topic.

## Figures and Tables

**Figure 1 ijerph-22-00501-f001:**
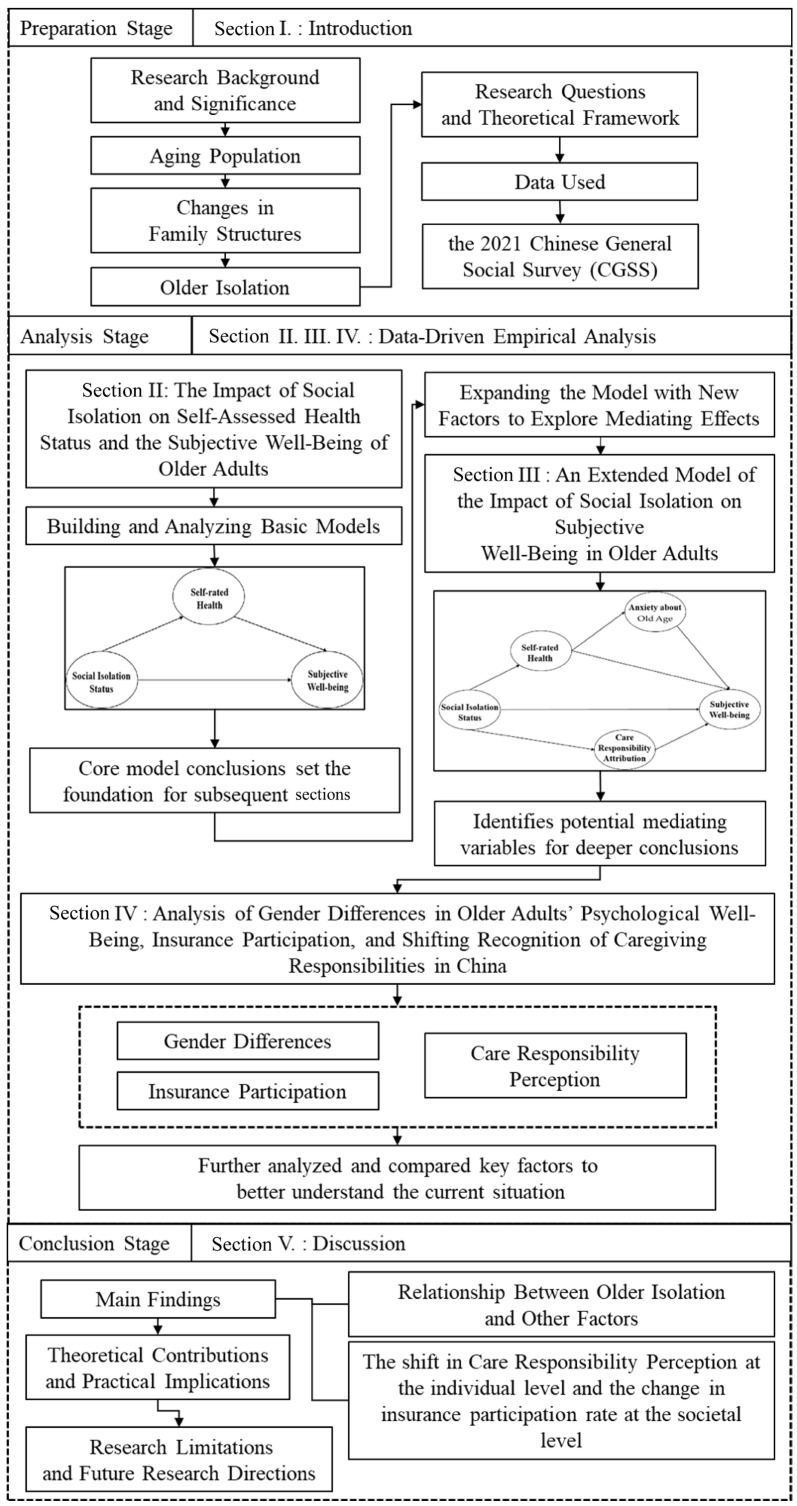
Structure and analytical method of the study.

**Figure 2 ijerph-22-00501-f002:**
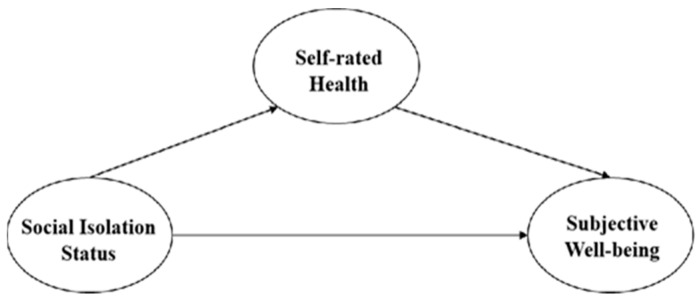
Estimation model.

**Figure 3 ijerph-22-00501-f003:**
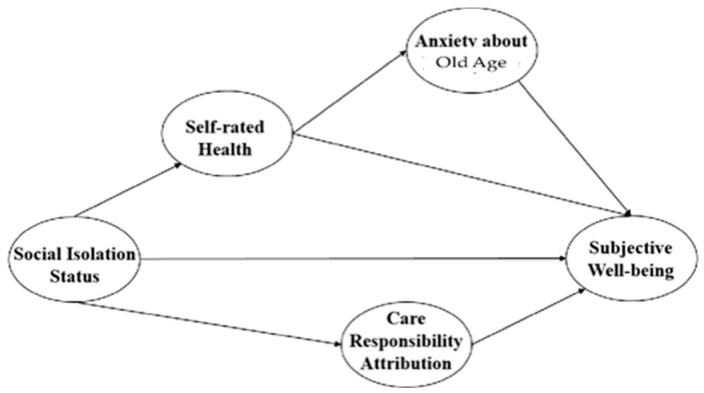
Extended estimation model.

**Figure 4 ijerph-22-00501-f004:**
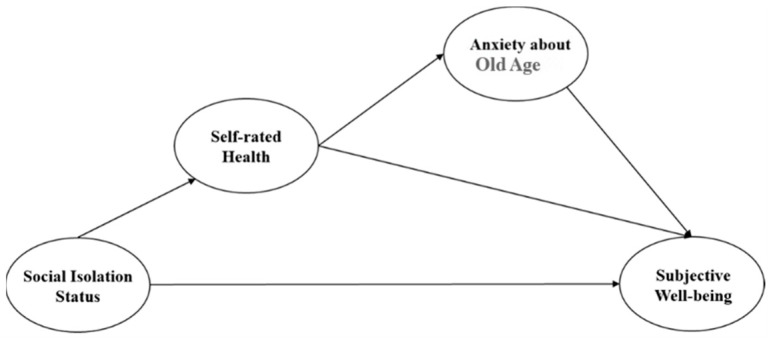
Final extended model.

**Table 1 ijerph-22-00501-t001:** Description of the variables.

Variable	Value
Social isolation status	Factor extraction conducted through exploratory factor analysis
Self-rated health	Overall, how do you feel about your current health status? 1 = Very Poor, 2 = Poor, 3 = Average, 4 = Good, 5 = Very Good
Subjective well-being	Overall, do you feel happy with your life? 1 = Very Unhappy, 2 = Unhappy, 3 = Average, 4 = Happy, 5 = Very Happy

**Table 2 ijerph-22-00501-t002:** Results of the analysis of the mediating effect of subjective well-being on the relationship between social isolation level and self-rated health.

C Total Effect	a	b	a × b Mediating Effect Value	a × b Boot SE	a × b z Value	a × b *p* Value	a × b 95% BootCI	c′ Direct Effect
−0.105 **	−0.191 **	0.135 **	−0.026	0.008	−3.086	0.002	−0.049~−0.016	−0.079 **

** *p* < 0.01. Percentile bootstrap method.

**Table 3 ijerph-22-00501-t003:** Additional variables and values.

Variable	Variable Values
Gender	Male = 0, Female = 1
Living situation	Living Alone = 0, Living Together = 1
Anxiety about old age	Factors extracted using exploratory factor analysis
Care responsibility attribution Who do you think should be primarily responsible for the care of older people with children? (Converted to dummy variables)	Rely primarily on children = 1 Rely primarily on government = 2 Self-responsibility = 3 Responsibility shared by government/children/self = 4
Insurance enrollment status	Urban/Rural Basic Medical Insurance, Participating = 1, Not Participating = 2 Urban/Rural Basic Pension Insurance, Participating = 1, Not Participating = 2 Private Medical Insurance, Participating = 1, Not Participating = 2 Private Pension Insurance, Participating = 1, Not Participating = 2

**Table 4 ijerph-22-00501-t004:** Results of multiple regression analysis for older and younger groups.

	Overall	Older	Younger (Comparison Group)
Constant	3.433 ** (26.143)	3.393 ** (11.527)	3.347 ** (22.634)
Living alone	−0.024 (−1.565)	−0.010 (−0.457)	−0.076 ** (−3.701)
Level of social isolation	−0.068 ** (−4.309)	−0.080 ** (−3.101)	−0.088 ** (−4.356)
Participation in private insurance	−0.049 (−1.695)	0.006 (0.088)	−0.098 ** (−3.173)
Anxiety about old age	0.103 ** (6.363)	0.116 ** (4.412)	0.078 ** (3.838)
Self-rated health	0.133 ** (9.025)	0.112 ** (4.598)	0.198 ** (10.369)
Sample Size	2717	1010	1707
R^2^	0.071	0.064	0.115
Adjusted R^2^	0.069	0.059	0.112
F value	F (5, 2711) = 41.430, *p* = 0.000	F (5, 1004) = 13.716, *p* = 0.000	F (5, 1701) = 44.240, *p* = 0.000

** *p* < 0.01.

**Table 5 ijerph-22-00501-t005:** Significance test of differences between the two regression models.

	Group 1	Group 2	Regression Coefficient b1	Regression Coefficient b2	Difference	t Value	*p* Value
Living situation	Older	Younger	−0.010	−0.076	0.066	2.166	0.030 *
Level of social isolation	Older	Younger	−0.080	−0.088	0.008	0.260	0.795
Anxiety about old age	Older	Younger	0.116	0.078	0.037	1.167	0.243
Self-rated health	Older	Younger	0.112	0.198	−0.085	−2.927	0.003 **
Participation in private insurance	Older	Younger	0.006	−0.098	0.104	1.422	0.155

* *p* < 0.05, ** *p* < 0.01.

**Table 6 ijerph-22-00501-t006:** Mediation effect analysis results for self-rated health ⟶ anxiety about old age ⟶ subjective well-being.

C Total Effect	a	B	a × b Mediation Effect Value	a × b Boot SE	a × b z Value	a × b *p* Value	a × b 95% BootCI	c′ Direct Effect	
0.107 **	0.066 **	0.097 **	0.006	0.003	2.167	0.030	0.004–0.018	0.100 **	

** *p* < 0.01. Bootstrap type = percentile bootstrap method.

**Table 7 ijerph-22-00501-t007:** Mediation effect analysis results for social isolation ⟶ care responsibility attribution ⟶ subjective well-being.

C Total Effect	a	b	a × b Mediation Effect Value	a × b Boot SE	a × b z Value	a × b *p* Value	a × b 95% BootCI	c′ Direct Effect
−0.105 **	−0.146 **	0.004	−0.001	0.004	−0.141	0.888	−0.008–0.008	−0.104 **

** *p* < 0.01. Bootstrap type = percentile bootstrap method.

**Table 8 ijerph-22-00501-t008:** Mediation effect analysis results for social isolation ⟶ self-rated health ⟶ anxiety about old age.

C Total Effect	a	b	a × b Mediation Effect Value	a × b Boot SE	a × b z Value	a × b *p* Value	a × b 95% BootCI	c′ Direct Effect
−0.036	−0.191 **	0.195 **	−0.037	0.008	−4.386	0.000	−0.055–−0.022	0.001

** *p* < 0.01. Bootstrap type = percentile bootstrap method.

**Table 9 ijerph-22-00501-t009:** Chi-square test results for insurance enrollment among older adults living alone (age ≥ 60).

Living Alone (Mean ± Standard Deviation)	Urban/Rural Basic Medical Insurance Enrollment	Urban/Rural Basic Pension Insurance Enrollment	Private Medical Insurance Enrollment	Private Pension Insurance Enrollment
−2.31 (*n* = 462)	0.09 ± 0.28	0.22 ± 0.41	0.96 ± 0.20	0.96 ± 0.20
0.43 (*n* = 2467)	0.06 ± 0.24	0.20 ± 0.40	0.94 ± 0.23	0.96 ± 0.19
F	4.329	1.291	1.292	0.139
*p*	0.038 *	0.256	0.256	0.709

* *p* < 0.05

**Table 10 ijerph-22-00501-t010:** Chi-square test results for insurance enrollment status among older/younger individuals.

Theme	Response	Older or Younger (%)		Total (%)	χ^2^	*p* Value
		Older (%)	Younger (%)			
Urban/rural basic medical insurance	Participating	2724 (93.00)	4889 (93.68)	7613 (93.43)	1.397	0.237
	Not Participating	205 (7.00)	330 (6.32)	535 (6.57)		
Urban/rural basic pension insurance	Participating	2319 (79.17)	3477 (66.62)	5796 (71.13)	143.948	0.000 **
	Not Participating	610 (20.83)	1742 (33.38)	2352 (28.87)		
Private medical insurance	Participating	154 (5.26)	962 (18.43)	1116 (13.70)	275.491	0.000 **
	Not Participating	2775 (94.74)	4257 (81.57)	7032 (86.30)		
Private pension insurance	Participating	107 (3.65)	454 (8.70)	561 (6.89)	74.507	0.000 **
	Not Participating	2822 (96.35)	4765 (91.30)	7587(93.11)		

** *p* < 0.01.

**Table 11 ijerph-22-00501-t011:** Chi-square test results for anxiety about old age, self-rated health, subjective well-being among male/female respondents.

Gender (Mean ± Standard Deviation)	Anxiety About Old Age	Self-Rated Health	Subjective Well-Being
Male (*n* = 1426)	2.74 ± 1.04	3.17 ± 1.11	4.11 ± 0.81
Female (*n* = 1503)	2.58 ± 1.00	2.98 ± 1.13	4.01 ± 0.90
T	2.423	4.626	2.651
*p* value	0.016 *	0.000 **	0.008 **

* *p* < 0.05, ** *p* < 0.01.

**Table 12 ijerph-22-00501-t012:** Chi-square test results for insurance enrollment among older adults living alone (age ≥ 60).

Rely on Government	Regression Coefficient	Standard Error	z Value	Wald χ^2^ Value	*p* Value	OR	OR 95% CI
Social isolation	−0.095	0.095	−0.998	0.997	0.318	0.909	0.755–1.096
Intercept	−1.341	0.101	−13.224	174.876	0.000	0.262	0.215–0.319
Self-responsibility	Regression Coefficient	Standard Error	z value	Wald χ^2^ Value	*p* value	OR	OR 95% CI
Social isolation	0.063	0.102	0.621	0.386	0.534	1.065	0.872–1.301
Intercept	−1.492	0.110	−13.562	183.939	0.000	0.225	0.181–0.279
Shared responsibility	Regression Coefficient	Standard Error	z value	Wald χ^2^ Value	*p* value	OR	OR 95% CI
Social isolation	−0.333	0.075	−4.413	19.477	0.000	0.717	0.618–0.831
Intercept	−0.718	0.080	−8.983	80.700	0.000	0.488	0.417–0.570

McFadden R^2^ = 0.009, Cox & Snell R^2^ = 0.022, Nagelkerke R^2^ = 0.025.

**Table 13 ijerph-22-00501-t013:** Analysis of variance results.

Care Responsibility Attribution (Mean ± Standard Deviation)	Subjective Well-Being
Rely primarily on children (*n* = 1453)	4.04 ± 0.71
Rely primarily on government (*n* = 380)	3.95 ± 0.79
Self-responsibility (*n* = 382)	4.10 ± 0.65
Responsibility shared among government/children/older adults (*n* = 714)	4.05 ± 0.68
F value	3.249
*p* value	0.021 *

* *p* < 0.05

## Data Availability

Restrictions apply to the availability of these data. Data were obtained from publicly available and are available at the Chinese National Survey Data Archive (http://www.cnsda.org/index.php; accessed on 21 August 2024).

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
