# Peer review of "The Impact of Social Isolation on the Subjective Well-Being of Older People in China: An Empirical Analysis Based on the 2021 China General Social Survey"

_ijerph, 2025, doi:10.3390/ijerph22040501_

Round 1
Reviewer 1 Report
Comments and Suggestions for Authors
This manuscript focuses on the psychological state and economic preparedness of socially isolated elderly individuals. It aims to delve into the determinants of subjective well-being among elderly individuals in isolation, thereby providing insights for improving elderly support resources and policies. On the whole, the topic is timely and relevant in terms of China’s increasing social challenges of population ageing, and such studies are of great valuable to provide academic insights and useful information for the development of elderly policies. However, my primary concern for this research is the fragmented research questions and content organization, which leads to the recommendation of reject.
The biggest issue is that this study tried to solve several research questions but lack of a clear overarching framework. Specifically, the first section tried to explore the relationships among social isolation, health and subjective wellbeing; afterwards, the second section examined the gender differences in subjective wellbeing, insurance participation and caregiving responsibility perceptions. Then questions come to the logical relationship between these two sections: why the gender difference is important? What’s the association of social isolation with the insurance participation? Why did you choose insurance participation and caregiving responsibility perceptions as the major explanatory factors that influence social isolation? Furthermore, the third section extends the model to include anxiety about old age and caregiving responsibility attribution as mediating variables. I doubt about the logic that develop the extended model from the simple model in terms of the original research aim of exploring the Impact of social isolation on subjective well-being. Although each section presents interesting findings, the overall organization and research design appear fragmented, and the connections between them are not well articulated. It seems that the manuscript is a compilation of separate studies rather than a cohesive investigation, making it difficult for readers to understand the cumulative contribution of the research.
Thus, I highly recommend the authors to focus on a single, well-defined research question and to construct a consistent methodological framework throughout. I believe that with substantial revisions and further development, the manuscript has the potential to make a valuable contribution to existing literature.
Reviewer 2 Report
Comments and Suggestions for Authors
The study presents clear hypotheses, it would benefit from further explanation on the rationale behind each hypothesis. For example, why do the authors expect social isolation to primarily affect subjective well-being through self-assessed health? Expanding on this connection would strengthen the theoretical framework. The paper relies on the 2021 China General Social Survey (CGSS) data, which is noted as a robust national dataset. However, the sample size of elderly individuals (1,010 respondents) may still present limitations in terms of generalizability. A discussion of potential biases such as the underrepresentation of certain rural areas or individuals with severe disabilities would enhance the credibility of the findings.
The study uses Principal Component Analysis (PCA) to create a composite variable for social isolation, which is a reasonable approach. However, the Cronbach’s alpha coefficient of 0.654 is on the lower end of the acceptable threshold. It would be helpful to further justify the reliability of this measure and possibly discuss any sensitivity analysis performed to confirm the robustness of the results. The analysis appears to consider gender differences, but more detailed exploration into how various socioeconomic factors (e.g., income, education, and employment status) influence subjective well-being would add depth. Specifically, how might financial insecurity or economic participation influence the elderly's experience of social
The paper mentions that social isolation alters elderly individuals' perception of caregiving responsibility. Expanding on this point with more detailed findings or examples would provide greater insight into the complexities of caregiving in isolated elderly populations. The paper offers valuable insights for improving elderly support systems. However, the paper could benefit from a more comprehensive set of policy recommendations or actionable insights based on the findings. For instance, specific interventions to mitigate social isolation or improve health care access for isolated elderly people could be discussed in greater detail.
The 2021 data used is relatively recent, but the authors mention that newer data might provide additional insights. If possible, incorporating more recent studies or data sources could further enhance the relevance and timeliness of the findings, especially given the rapidly changing social landscape in China.
The use of Chi-square tests, t-tests, and regression analysis is appropriate for testing the hypotheses. However, more justification for the choice of these tests would improve the methodological rigor. For instance, the paper mentions a "multinomial logistic regression" but could clarify why this method is the most suitable for analyzing the perceptions of caregiving responsibilities. A brief explanation of why each statistical test was chosen in relation to the variables and hypotheses would add depth to the methodology section. The data tables, while informative, could be improved by providing more context for interpretation. For example, Table 3-2 and Table 3-4 present Chi-square test results but could benefit from additional guidance on how to interpret p-values and effect sizes. In the tables, explicitly point out the significance of findings and discuss their implications briefly within the text. For instance, when referencing the differences in anxiety about old age between genders, highlight not only the p-value but also the practical implications of the results.
The results are generally well presented, the regression results on caregiving responsibility perceptions (H1) require more detailed discussion. The relationship between social isolation and caregiving perceptions is complex and warrants deeper analysis. The lack of significant findings for "relying on the government" and "self-responsibility" should be discussed more thoroughly in the context of the literature. Address the unexpected findings in greater detail. Why might social isolation not significantly affect perceptions of caregiving responsibility in some categories? Exploring possible reasons for this, such as cultural factors or socioeconomic variables, could provide more nuanced insights.
The paper provides a solid foundation for policy implications, especially concerning the role of gender in elderly care. However, the implications for social security and insurance systems in China could be more fully explored. Provide more specific policy recommendations based on the findings. For example, if gender differences in anxiety and well-being are significant, suggest ways that insurance systems or caregiving policies could be tailored to address these differences.
Reviewer 3 Report
Comments and Suggestions for Authors
Comments:
1. This paper analyzes the impact of social isolation on subjective well-being of the elderly in China. The reported data of Shanghai mentioned in the first paragraph of the introduction is obviously not very appropriate.
2. The analysis of the elderly insurance participation is shallow, and the factors such as regional and urban-rural differences are not fully considered.
2. Is the object of this study all elderly people or those who live alone? The title is Chinese elderly people, but the third paragraph of the introduction also mentions elderly people living alone. It is suggested to clarify the research object and further clarify in the method section.
3. Although the existing data provide basic information, it lacks a deep understanding of the actual situation of the participants, and further research should be combined with qualitative methods.
4. The structure of the paper is suggested to be re-adjusted. The method section includes data sources and variable measurements, and how to measure all used variables is placed in the method section. The hypothesis part is separated from the result part. The article follows "1. Introduction, 2. Theoretical hypotheses, 3. Methods, 4. Conclusions, 5. Discussion "structure to organize content.
Round 2
Reviewer 2 Report
Comments and Suggestions for Authors
Dear Authors,
I am pleased with the corrections and revisions made by the authors in response to the feedback provided. The authors have effectively addressed the concerns raised, demonstrating a clear understanding of the issues and a commitment to improving the manuscript's quality.
Overall, I am satisfied with the corrections.
Thank you.
Comments on the Quality of English Language
The English could be improved to more clearly express the research.
Author Response
Dear Reviewer,
Thank you for your valuable review and feedback. We sincerely appreciate your efforts and acknowledge the areas for improvement in our use of English. To enhance the clarity and quality of our manuscript, we are currently utilizing Elsevier's language editing service. We are committed to further refining the manuscript and ensuring its readability.
Once again, thank you for your time and constructive comments.
Best regards,
Ryu Keikoh, Chen Zaiqing